# Systemic Analyses of Cuproptosis-Related lncRNAs in Pancreatic Adenocarcinoma, with a Focus on the Molecular Mechanism of LINC00853

**DOI:** 10.3390/ijms24097923

**Published:** 2023-04-27

**Authors:** Leifeng Chen, Lin Zhang, Haihua He, Fei Shao, Yibo Gao, Jie He

**Affiliations:** 1Department of Oncology, Renmin Hospital of Wuhan University, Wuhan 430060, China; 18770099029@126.com (L.C.); zhanglinwhu@foxmail.com (L.Z.); 2019283020072@whu.edu.cn (H.H.); 2Department of Thoracic Surgery, National Cancer Center/National Clinical Research Center for Cancer/Cancer Hospital, Chinese Academy of Medical Sciences and Peking Union Medical College, Beijing 100021, China; shmf2009@live.cn; 3Laboratory of Translational Medicine, National Cancer Center/National Clinical Research Center for Cancer/Cancer Hospital, Chinese Academy of Medical Sciences and Peking Union Medical College, Beijing 100021, China; 4Central Laboratory & Shenzhen Key Laboratory of Epigenetics and Precision Medicine for Cancers, National Cancer Center/National Clinical Research Center for Cancer/Cancer Hospital & Shenzhen Hospital, Chinese Academy of Medical Sciences and Peking Union Medical College, Shenzhen 518116, China; 5State Key Laboratory of Molecular Oncology, National Cancer Center/National Clinical Research Center for Cancer/Cancer Hospital, Chinese Academy of Medical Sciences and Peking Union Medical College, Beijing 100021, China

**Keywords:** pancreatic cancer, lncRNA LINC00853, aerobic glycolysis, prognostic signature, cuproptosis

## Abstract

Pancreatic cancer (PC) is a deadly malignant digestive tumor with poor prognoses and a lack of effective treatment options. Cuproptosis, a recently identified copper-dependent programmed cell death type, has been implicated in multiple cancers. Long non-coding RNAs (lncRNAs) are also linked to the progression of PC. However, the role and prognostic values of cuproptosis-related lncRNAs in pancreatic adenocarcinoma (PAAD) remain unclear. In this study, we systemically analyzed the differential expressions and prognostic values of 672 cuproptosis-related lncRNAs in PAAD. Based on this, a prognostic signature including four lncRNAs (*LINC00853*, *AC099850.3*, *AC010719.1*, and *AC006504.7*) was constructed and was able to divide PAAD patients into high- and low-risk groups with significantly different prognoses. Next, we focused on lncRNA *LINC00853*. The differential expressions of LINC00853 between normal tissue and PAAD samples were validated by qRT-PCR. LINC00853 was knocked down by siRNA in PC cell lines BxPC-3 and PANC-1 and the oncogenic role of LINC00853 was validated by CCK8, colony formation, and EdU assays. Subsequently, LINC00853 knockdown cells were subjected to tumor xenograft tests and exhibited decreased tumor growth in nude mice. Mechanistically, knockdown of LINC00853 significantly reduced cellular glycolysis and enhanced cellular mitochondrial respiration levels in PC cells. Moreover, knockdown of LINC00853 decreased the protein level of a glycolytic kinase PFKFB3. Finally, glycolysis tests and functional tests using LINC00853 and HA-PFKFB3 indicated that the effects of LINC00853 on glycolysis and cell proliferation were mediated by PFKFB3. In conclusion, our systemic analyses have highlighted the important roles of cuproptosis-related lncRNAs in PAAD while the prognostic signature based on them showed excellent performance in PAAD patients and is expected to provide clinical guidance for individualized treatment. In addition, our findings provide a novel mechanism by which the LINC00853-PFKFB3 axis critically regulates aerobic glycolysis and cell proliferation in PC cells.

## 1. Introduction

As a prevalent malignant tumor of the digestive system, pancreatic cancer (PC) is the fourth primary cause of death associated with cancer worldwide, showing an extremely poor prognosis with high mortality among these patients [1,2]. Approximately 90% of tumor tissues are derived from the pancreatic ductal epithelium; thus, pancreatic ductal adenocarcinoma (PDAC) is the most prevalent pathological type [3]. Although great progress has been made in the prevention, diagnosis, and treatment of PDAC, the overall five-year survival rate of patients with PC is unfortunately less than 9%; particularly, the median survival time of patients with PDAC is less than 7.8 months in China [4]. The overall efficacy and survival benefit of current medical management of PAAD remain worrisome and may be largely attributed to the fact that only 15–20% of patients are diagnosed at a stage that has surgical indication and most are characterized by metastasis even at an early stage [5,6]. Therefore, in-depth clarification of the basic mechanism of PAAD tumorigenesis and identification of reliable and specific biomarkers for its diagnosis and evaluation of prognosis are necessary for improving the efficiency of early PAAD diagnosis and developing new therapeutic strategies for these patients.

Existing literature suggests that in cells, multiple kinds of precisely regulated cell death (RCD) types operate, including apoptosis, ferroptosis, necrosis, and thermal apoptosis; these subroutines of RCD differ in terms of the initiating stimuli, intermediate activation events, and final effectors [7,8]. Heavy metal ions are essential trace elements in cells [9]; under normal organismal physiological conditions, these are usually maintained at low concentrations in a dynamic balance. However, when the metal element content is insufficient or excessive, cell death is triggered [10,11,12]. For instance, ferroptosis, an oxidative iron-dependent cell death, results from an unlimited lipid peroxidation [13]. Like iron, copper is an intracellular trace metal required for several biological processes, including the synthesis of biological compounds, antioxidant defense, and mitochondrial respiration; copper excess can lead to cytotoxicity [14,15,16]. Surprisingly, copper ions can cause cell death even when the known cell death pattern is blocked and this is therefore defined as cuproptosis. This cell death type depends on copper and is triggered through the direct binding of copper ions with lipid-acylated components of the tricarboxylic acid cycle (TCA) during mitochondrial respiration. This results in the accumulation of lipid-acylated proteins and concomitant reduction in iron–sulfur cluster proteins, causing proteotoxic stress and eventually cell death [17,18]. Ten key genes that regulate cuproptosis have been identified. The knockdown of seven cuproptosis regulate genes (*FDX1*, *LIPT1*, *LIAS*, *DLD*, *DLAT*, *PDHA1*, and *PDHB*) rescued the cytotoxic effects of elesclomol and diethyldithiocarbamate. These seven cuproptosis regulated genes (CRGs) are positively regulated during cuproptosis, and MTF1, GLS, and CDKN2A are negatively regulated during cuproptosis [17]. Several links have been observed between copper and cancer. Copper accumulation is closely associated with tumor cell development, angiogenesis, and metastasis [12,19,20,21]. These findings have inspired research on the physiological normalization and alteration of mitochondrial copper homeostasis and its use as a cancer therapeutic target.

Long non-coding RNAs (lncRNAs) are a subtype of ncRNAs comprising more than 200 nucleotides [22]. Research suggests that lncRNAs participate in the malignant progression of several cancers, including PC, at transcriptional, post-transcriptional, and epigenetic levels. These are regarded as specific and sensitive cancer biomarkers for the diagnosis, treatment, and prognosis of PAAD [23,24]. For instance, Zhang et al. [25]. found that the lncRNA PSMB8-AS1 facilitated the development of PC by modulating the PD-L1/STAT1/miR-382-3P axis. Additionally, Huang et al. [26]. indicated that in PC tissues, LINC00842 had a high expression, which was evidently related to the patient’s poor prognosis. Further evaluation revealed that LINC00842 caused metabolic reprogramming of PC cells through its interaction with the transcription cofactor, PGC-1α, thereby promoting its malignant progression. However, to date, the association of cuproptosis-associated lncRNA with the prognosis of PAAD has not been reported. Thus, this study aimed at determining and confirming a new multi-lncRNA diagnostic signature associated with cuproptosis to precisely predict the survival of PAAD patients and provide valuable information for individual management and clinical decision-making for these patients.

## 2. Results

### 2.1. Determination of Differentially Expressed Cuproptosis-Related lncRNAs in the PAAD Cohort

Figure 1 displays the flow chart of the study. Comprehensive analysis of the transcriptomic data of 171 normal pancreatic tissues together with 178 PC tissues from acquired GTEx and TCGA databases, respectively, yielded 672 lncRNAs (*r* > 0.4 and *p* < 0.001) co-expressed with 10 cuproptosis-related regulatory genes (Figure 2A, Appendix A). These lncRNAs were analyzed for the differential expression between tumor and normal samples (*p* < 0.05 and |log_2_ fold change| > 2), and 91 cuproptosis-related lncRNAs were obtained. Among them, the expressions of 50 were attenuated, while those of 41 were enhanced (Figure 2B,C). Appendix A displays a network diagram for the lncRNAs and cuproptosis-associated regulatory genes.

### 2.2. Construction and Verification of Prognostic Signature According to Cuproptosis-Related lncRNAs

Univariate Cox regression analysis indicated a significant correlation (all *p* < 0.05) between the 15 cuproptosis-associated lncRNAs and the overall survival (OS) of patients with PAAD (Figure 3A). A heatmap was plotted to reveal the differential expression of lncRNAs between PC and normal pancreatic tissues (Appendix A). To avoid the over-fitting of prognostic features, LASSO regression analysis was performed based on the 15 chosen lncRNAs related to the prognosis of PAAD patients, and four cuproptosis-associated lncRNAs were acquired following minimum partial likelihood deviation, including *LINC00853*, *AC099850.3*, *AC010719.1*, and *AC006504.7* (Figure 3B and Appendix A). For in-depth optimization of the outcomes, based on the multivariate Cox regression coefficients and lncRNA expression, a prognostic signature for PAAD was established. For the cuproptosis-associated lncRNAs, prognostic signature scores were computed as follows: (*LINC00853* × 1.334352 expression) + (*AC099850.3* × 0.794074 expression) − (*AC010719.1* × 0.76659 expression) − (*AC006504.7* × 1.518996 expression). Among them, significant risk genes were *AC099850.3* and *LINC00853*, while the protective genes were *AC010719.1* and *AC006504.7* (Table 1). Next, patients in training and test, as well as the whole TCGA-PAAD cohort were classified into high- and low-risk groups according to the median risk score (Appendix A). Table 2 reveals the clinical features of the test (*n* = 89) and TCGA training (*n* = 88) groups. t-SNE analysis and PCA confirmed the clustering ability of the risk score according to four cuproptosis-associated lncRNAs in the PAAD patient cohort, clearly distinguishing patients between different subgroups (Appendix A). The distribution of risk score and survival status of patients in the above cohort showed that the high-risk group had more deaths and shorter survival duration (Figure 4A–F).

The heatmaps for the four cuproptosis-related lncRNAs are shown in Figure 4G–I. The levels of AC099850.3 and LINC00853 expression were enhanced in the high-risk group, while those of AC010719.1 and AC006504.7 decreased significantly. Notably, survival analyses in the training set exhibited that the low-risk group had significantly longer OS and PFS in contrast to the high-risk group (*p* < 0.01, Figure 4J,M). As expected, the whole TCGA-PAAD cohort and test cohort showed similar outcomes (Figure 4K,L,N,O). In the training cohort, the ROC curve exhibited AUC values of 1-, 3-, and 5-year survival at 0.804, 0.761, and 0.888, respectively, indirectly reflecting the robust predictive ability of the cuproptosis-related multi-lncRNA prognostic signature herein (Figure 4P–R).

### 2.3. Subgroup Analysis of the Prognostic Value of the Cuproptosis-Related Multi-lncRNA Signature

As shown in Appendix A, we first examined the distribution of routine clinicopathological characteristics between the risk groups classified based on the cuproptosis-related lncRNA signature, wherein the tumor grade (*p* < 0.01) differed significantly between the two groups. To assess whether the prognostic value of multi-lncRNA signature associated with cuproptosis was independent of conventional clinicopathological features, PAAD patients were classified into various subgroups according to clinicopathological features, including sex, age, stage, and tumor grade. Young (≤60 years, *p* < 0.001) (Appendix A) or old patients (>60 years, *p* < 0. 001) (Appendix A), male (*p* = 0.005) (Appendix A) or female patients (*p* < 0.001) (Appendix A), low (grade 1–2, *p* < 0.001) (Appendix A) or higher tumor grade (grade 3–4, *p* = 0.005) (Appendix A), and stages I–II (*p* < 0.001) (Appendix A) showed statistical significance following Kaplan–Meier survival curve analysis. Patient sample size in stages III–IV (*p* = 0.134) (Appendix A) was small which may have resulted in an insignificant statistical difference between the two groups. These results suggested that the risk score-based multi-lncRNA signature associated with cuproptosis was a powerful tool for the prediction of PAAD survival in different clinical subgroups divided by age, stage, tumor grade, and sex.

Subsequently, multivariate and univariate Cox regression analyses were performed to determine whether the cuproptosis-related multiple-lncRNA signature possessed a prognostic value independent of the clinicopathological indicators. Following univariate Cox regression analysis for TCGA-PAAD, the patient’s risk score and tumor grade separately predicted poor survival (risk score: HR = 1.098, 95% CI: 1.041–1.158, *p* < 0.001; grade: HR = 1.383, 95% CI: 1.019–1.877, *p* = 0.037, Figure 5A). Multivariate Cox regression analysis in TCGA-PAAD showed that after the inclusion and correction of other confounders, the patient’s risk score and tumor grade were independent predictors of OS (risk score: HR = 1.093, 95% CI = 1.041–1.149, *p* < 0.001; grade: HR = 1.383, 95% CI: 1.035–1.847, *p* = 0.028, Figure 5B).

### 2.4. Somatic Mutational Landscape and Drug Sensitivity Analysis

According to the somatic mutation data from the TCGA-PAAD cohort, the 20 most commonly mutated genes in PAAD patients were analyzed (Appendix A). Afterwards, the differences in the somatic mutations between the two groups were investigated (Figure 6A,B). Mutational profile features suggested that missense mutations were the most prevalent in both groups and that somatic mutations in the high-risk group were more common compared to the low-risk group. The horizontal histogram on the right shows the frequency of mutations in the 20 aforementioned genes in each subgroup, with mutations in *KRAS* (76 vs. 45%), *TP53* (69 vs. 41%), *CDKN2A* (25 vs. 11%), *MUC16* (11 vs. 3%), and *TGFBR2* (9 vs. 0%) differing the most between the two patient groups. Mutations in *KRAS* and *p53* were the most prevalent in patients of the two groups. Subsequently, the correlation of prognostic characteristics with TMB was assessed. The outcomes suggested that in contrast to patients in the low-risk group, those in the high-risk group had evidently higher TMB (Figure 6C). Eventually, the difference in sensitivity to the chemotherapeutic agents between low- and high-risk patients was estimated using the GDSC database and the pRRophetic R package. As shown in Figure 6D,F–O, in the low-risk group, these patients were more sensitive to multiple chemotherapeutic agents, including inhibitors of AKT (MK.2206), Bcl2 (TW.37), FGF-VEGF receptor tyrosine kinase (PD.173074), HDAC (vorinostat), MDM2 (Nutlin.3a), mTOR (AZD8055), WIP1 (CCT007093), and Rac family small GTPases (EHT.1864), as well as palbociclib, motesanib, and axitinib. In high-risk groups, the patients were more sensitive to PLK inhibitors (BI. 2536) (Figure 6E).

### 2.5. Analysis of Tumor Immune Microenvironment and Pathways Related to Prognostic Features

The acquisition of stem and progenitor cell-like characteristics and progressive loss of differentiation phenotype are the critical features of the malignant development of tumor cells [27]. Nonetheless, together with the recurrence of the malignant tumor cells, the tumor stem cell-like characteristics are the primary cause of unlimited proliferation. These characteristics can be determined using the DNA dryness score (DNAss) based on the DNA methylation modes and RNA dryness score (RNAss) according to the mRNA expression [28]. The tumor microenvironment is thought to exert crucial effects on tumor progression, recurrence, metastasis, as well as drug resistance [29]. The correlation between risk scores and tumor dryness in PAAD patients was analyzed. Outcomes displayed an evident association between risk scores and DNAss (*R* = 0.18, *p* = 0.026) and RNAss (*R* = 0.31, *p* = 9.1 × 10^−5^) (Figure 7A). ssGSEA was performed to analyze the differences in immune cell infiltration features in the tumor microenvironment between both groups to count the relative fraction of 28 immune cells in each PAAD patient. Notably, in the low-risk group, the activated CD56dim NK cells, CD4 T cells, Th2 cells as well as neutrophils were abundant (Figure 7B). Immune checkpoint gene expression analysis indicated an evident difference between the risk groups. For instance, compared to the high-risk group, the low-risk group had higher expressions of CD200, ADORA2A, TNFRSF4, TNFSF14, CD160, and TNFRSF25, while those of CD276, CD80, TNFSF18, TNFSF9, CD274, TNFSF4, and HHLA2 were enhanced in the high-risk group (Figure 7C). Finally, we assessed the Spearman association of immune infiltration with the risk scores (Figure 7D), and the detailed results are provided in Appendix A. These findings suggested that the low-risk group characterized by immune response activation may contribute to anti-tumor effects. To elucidate the underlying biological pathways for the risk subgroups, KEGG pathway enrichment analysis was performed following the differential expression analysis for the coding genes between the high-risk and low-risk groups. Various tumor-related pathways were enriched. Moreover, pathways involved in the cell cycle, PI3K-Akt signaling, ECM-receptor interaction, P53_signaling_pathway, and regulation of actin cytoskeleton were significantly enriched (Figure 7E). These findings reflected a remarkable modulation of cuproptosis in the high-risk score subgroup and multiple signaling pathways may be involved.

### 2.6. LINC00853 Is an Oncogene Candidate Gene for PC Progression

Since the difference in the expression of LINC00853 was the most important lncRNA for predicting poor prognosis based on regression coefficients, we focused on LINC00853. The expression of LINC00853 was elevated in the PC tissues (178 PC tissues from TCGA cohort compared to 171 normal pancreatic tissues) (Figure 8A and Appendix A). Compared to the high LINC00853 group, PC patients in the low LINC00853 group showed better prognoses (Figure 8B). The qRT-PCR results were consistent with those of bioinformatics analysis, suggesting that the expression of LINC00853 was high in PC tissues compared to the corresponding paraneoplastic tissues (Figure 8C; *n* = 32). Additionally, to investigate the function of LINC00853 in PC, it was knocked down in two PC cell lines, BxPC-3 and PANC-1, using siRNAs (Figure 8D,E). As shown in Figure 8F,G, CCK-8 analysis suggested that PC cell growth was inhibited significantly following the silencing of LINC00853 in vitro. Consistent with these results, colony formation and EdU assays showed that the proliferation rate of BxPC-3 and PANC-1 cells reduced after knocking down LINC00853 (Figure 8H–K). Finally, we subjected cells with knocked-down LINC00853 to tumor xenograft tests. After five weeks of growth, LINC00853-blocked PANC-1 cells (shLINC00853/PANC-1) and exhibited decreased tumor growth in the nude mice in contrast to the controls. Similarly, the mean tumor weight of mice xenografted with shLINC00853/PANC-1 cells was remarkably attenuated (Figure 8L). Collectively, these data suggested that LINC00853 was conducive to PC cell growth in vivo and in vitro.

### 2.7. LINC00853 Enhances Aerobic Glycolysis and Proliferation through PFKFB3 in PC Cells

To elucidate the mechanism of LINC00853-mediated proliferation of PC cells, we performed GSEA in the TCGA cohort to assess the possible associations between LINC00853 and various signaling pathways. As shown in Figure 9A, the gene sets of glycolysis_targets were enriched in PC samples with high LINC00853 expression, indicating that the Warburg effect was closely associated with a high LINC00853 level in PC. Next, we evaluated the function of LINC00853 in glucose metabolism in PC cells. As shown in Figure 9B,C, knocking down LINC00853 reduced the levels of glucose-6-phosphate (G6P), glucose consumption, lactate production, and ATP in BxPC-3 and PANC-1 cells (Figure 9D–G). To verify the effect of LINC00853 on glycolysis in PC cells, we measured ECAR, reflecting the total glycolytic flux. Knocking down LINC00853 significantly reduced ECAR in BxPC-3 and PANC-1 cells. Cellular OCR, reflective of mitochondrial respiration, was also examined. shLINC00853/BxPC-3 and shLINC00853/PANC-1 cells showed enhanced OCR (Figure 9H–K). Our previous study confirmed that PFKFB3 showed high activity of glycolytic kinases, and its abnormal expression has been reported in many tumors [30]. To verify whether it was also altered in PC, the level of PFKFB3 was analyzed in LINC00853 knocked-down cells. PFKFB3 expression was found to be significantly reduced under these conditions (Figure 10A–D). Finally, we sought to verify whether the effects of LINC00853 on glycolysis and cell proliferation were mediated by PFKFB3. Initially, PANC-1 cells carrying shLINC00853 constructs were transfected with HA-PFKFB3. Colony formation and EdU assays demonstrated that PANC-1 cell proliferation was partially restored following PFKFB3 overexpression (Figure 10E–H). G6P, glucose consumption, and lactate production were partially restored in PANC-1 cells following transfection with HA-PFKFB3 (Figure 10I,J). In conclusion, these findings suggested that LINC00853 enhanced aerobic glycolysis and proliferation through PFKFB3 in PC cells.

## 3. Discussion

PC is a type of solid tumor with extremely high malignancy and poor patient prognosis. Treatment options and effects are still limited [2]. Currently, the lack of molecules to precisely target tumors for treatment and effective tumor-killing initiators poses a significant barrier to the progression of PAAD precision therapy [6]. However, recent research suggests that regulating the process of programmed tumor cell death can effectively improve the efficacy of targeted therapy for tumors [31,32,33]. Cuproptosis, a recently identified mechanism of programmed cell death, has attracted great interest in the field of carcinogenesis and anti-tumor treatment [12,17,34]. The process of cuproptosis depends primarily on the intracellular copper ion accumulation, which directly binds to the lipid-acylated components of the TCA cycle, causing the dysregulation and aggregation of proteins and the blockade of the TCA cycle, triggering proteotoxic stress, and inducing cell death [18]. However, most current studies on tumor cell cuproptosis have focused on the level of encoded proteins, whereas ncRNAs that can diversely affect tumorigenesis and progression have received insufficient attention. Numerous studies have indicated that localization and aberrant expression of several lncRNAs in cancer are significant factors underlying tumor development, and these participate in modulating programmed tumor death by binding to proteins, miRNAs, and DNA, thereby influencing the prognosis of patients with tumors and their therapeutic outcomes [35]. As a result, lncRNAs are gradually being acknowledged as new prognostic diagnostic markers for molecular-targeted therapies for human cancers. Moreover, our findings showed that prognoses and tumor treatment responses varied among patients with different PC subtypes and clinical characteristics [36]. Therefore, establishing robust prognostic assessment tools for treatment outcome prediction may better guide effective and personalized clinical intervention decisions for patients with PAAD. Herein, we constructed a reliable prognostic signature based on cuproptosis-associated lncRNAs and demonstrated its clinical utility for patients with PC. We preliminarily validated the lncRNA expression profiles based on the prognostic signature in PAAD, further confirming its reliability.

Furthermore, cellular cuproptosis and lncRNAs in PC were associated and we investigated the differential expression of cuproptosis-associated lncRNAs between normal pancreatic tissue and the PAAD samples. The prognostic utility of the expression of lncRNAs associated with cuproptosis in PAAD was assessed. Moreover, we determined and confirmed new prognostic signatures for predicting the immune responses together with the prognosis of PAAD patients according to the expression of differentially expressed and prognostically valuable lncRNAs associated with cuproptosis by LASSO Cox regression analysis and other bioinformatics approaches. According to the prognostic features, the PAAD patients were classified into low- and high-risk subgroups. Kaplan–Meier survival analysis showed that PFS and OS were worse in the high-risk group compared to the low-risk group. Multivariate and univariate Cox regression analyses, employed to assess the risk characteristics, revealed that cuproptosis-associated multi-lncRNA prognostic features were an independent prognostic indicator for PAAD patients in the whole, training, and test cohorts. The 1-, 3-, and 5-year survival ROC curves validated the predictive accuracy of the cuproptosis-related multi-lncRNA prognostic signature. Sex, age, grade, and clinical stage in the prognostic signature were effective except for the subgroup of stages III–IV, indicating the generalizability of this prognostic signature.

Immunotherapy is successful in rendering cancer curable by enhancing the immune system of these patients [37]. A large body of preclinical and clinical literature highlights that immune-based therapeutic strategies may provide survival benefits for patients with PAAD [38]. The combination of immunotherapy and other therapeutic approaches is likely to serve as an alternative option for PAAD treatment. Therefore, we analyzed the correlation between the tumor microenvironment and multi-lncRNA prognostic signature associated with cuproptosis in PAAD. ssGSEA was performed to assess the immune cell infiltrate abundances between both risk groups, which revealed that the activated cd56dim NK cells, CD4 T cells, Th2 cells, and neutrophils differed significantly. Similarly, we observed higher immune checkpoint marker expression (e.g., CD200, ADORA2A, TNFRSF4, TNFSF14 AND CD160, and TNFRSF25) in the high-risk group. Choueiry et al. [39] reported that the expression of CD200 in the PAAD microenvironment may modulate the expansion of myeloid-derived inhibitor cells and targeting CD200 may enhance the checkpoint activity in immunotherapy. Ma et al. [40] showed that a combination of OX40 (TNFRSF4) agonist and PD-1 suppressor resulted in immune memory and tumor rejection in a mouse model of PC. Combining this experimental evidence with our results suggested better outcomes for high-risk group patients receiving immune checkpoint suppressor treatments. The close association between tumor immunotherapy and lncRNAs has been elucidated in several studies [41,42]. The current report is the first to initially determine the underlying contribution of lncRNAs (LINC00853, AC099850.3, AC010719.1, and AC006504.7) in immunotherapy. Future research should validate these lncRNA functions in immunotherapy and focus on the detailed mechanisms by which they affect immunotherapy, particularly in PAAD. Moreover, we investigated the molecular mechanisms underlying cuproptosis-related lncRNAs by GSEA. The enrichment results also implicated other tumor pathways, suggesting that these lncRNAs and the prognostic signature derived from them may be equally applicable to other tumors, warranting further research. Furthermore, pathways related to tumors, like glycolysis, cell cycle, gluconeogenesis, and p53 signaling were enriched, indicating that cuproptosis-related lncRNAs may contribute to the development of PAAD through these pathways.

Overall gene mutations and TMB in various risk groups were assessed. TMB refers to the total somatic mutation number per megabase of the interrogated genomic sequence. High TMB in tumor cells results in more immunogenic neoantigens, while the identification of neoantigens through host T cells (particularly T cytotoxic lymphocytes) is one of the most significant aspects underlying the prediction of immunotherapeutic responses [43]. Recent research indicates that TMB is a biomarker for immunotherapeutic response in PC [44,45]. Our outcomes displayed that the variables of overall mutation differed between both risk groups, with the high-risk group showing a higher TMB than the low-risk group, consistent with the findings of the tumor microenvironment analysis, revealing that high-risk group patients may gain more benefit from immunotherapy. Different clinical trials have demonstrated the robustness of utilizing the GDSC database and the “pRRophetic” R package in predicting responses to chemotherapy [46]. The IC_50_ values of various targeted drugs were computed to evaluate the sensitivity of patients to these compounds, and the two subgroups showed respective relative sensitivity to the targeted agents. These could be potential compounds for treating PAAD and may target specific cellular cuproptosis-related lncRNAs. Collectively, these findings may provide more suitable personalized treatment options for patients with PAAD. We anticipate that clinical trials will be conducted subsequently to confirm our findings. Moreover, we detected the lncRNA expression based on the cuproptosis-related multi-lncRNA prognostic signature in PC tissues and cells by qRT-PCR, and the outcomes were consistent with those of bioinformatics analysis using datasets, suggesting that prognostic characteristics are promising tools for the prediction of survival in PAAD patients.

Enhanced glycolysis is strongly associated with cancer progression and is related to poor prognosis [47]. Targeting glycolytic metabolism in cancer is a novel treatment approach for cancer [48]. Herein, cuproptosis-related lncRNA LINC00853 was found to play a key role in regulating the glycolysis of PC cells. We revealed a novel mechanism by which LINC00853 regulated glycolysis by affecting PFKFB3 expression. This conclusion was based on the following observations. First, the proliferation rate of PC cells was reduced after the knockdown of LINC00853. Second, LINC00853 was conducive to PC cell growth in vivo. Third, knocking down LINC00853 reduced the levels of G6P, glucose consumption, lactate production, and ATP in BxPC-3 and PANC-1 cells. Finally, PC cell proliferation was partially restored by PFKFB3 overexpression. G6P, glucose consumption, and lactate production were partially restored in PC cells following HA-PFKFB3 transfection. Taken together, these data demonstrated that LINC00853 regulated PC cell proliferation and aerobic glycolysis via a mechanism dependent on PFKFB3.

Nevertheless, our study has some limitations that warrant consideration. First, the cohort in this paper was primarily based on the TCGA database. Therefore, different databases and large multicenter cohorts are required for in-depth external verification of the cuproptosis-related multi-lncRNA prognostic signature. Second, only the expression of lncRNAs in the signature was preliminarily validated in this study, and we did not conduct a detailed investigation of the exact roles and mechanisms of action in PC cells or the tumor microenvironment and cuproptosis of PC. This warrants further in vitro and in vivo experimental validation.

In conclusion, we established a cuproptosis-associated multi-lncRNA prognostic signature for predicting the prognosis of PAAD patients. The signature score could be used as an independent prognostic indicator and was associated with the tumor immune infiltration level and effectiveness of tumor immunotherapy. This approach may provide a theoretical basis for enhancing the anti-tumor immune efficacy in PAAD and developing novel therapeutic strategies. The findings are of significance for future basic research and clinical work.

## 4. Materials and Methods

### 4.1. Acquisition and Processing of Data from PAAD Patients

From The Cancer Genome Atlas (TCGA; https://portal.gdc.cancer.gov/, accessed on 31 March 2022) (containing 178 PC tissues and 4 adjacent pancreatic tissues) and Genotype-Tissue Expression Project (GTEx; https://www.gtexportal.org/, accessed on 31 March 2022) (including 167 normal pancreatic tissues) databases, the normalized RNA-seq data were downloaded. Human gene transfer format (GTF) files for annotating gene IDs to extract all mRNA and lncRNA expression matrices were downloaded from the Ensembl (http://asia.ensembl.org, accessed on 31 March 2022) dataset. Clinical and mutation data (VarScan version) of 185 and 158 PAAD patients, respectively, were similarly obtained from the TCGA database. This study followed the publication guideline requirements of the respective databases.

### 4.2. Determination of Differentially Expressed Cuproptosis-Associated lncRNAs with Prognostic Value

Ten key genes reportedly regulating cuproptosis were found to be differentially expressed in PC (Appendix A). According to gene annotation, the lncRNAs in the TCGA cohort were screened, and 13365 lncRNAs were acquired. By calculating the Pearson correlation coefficient, the association between lncRNAs and the 10 genes related to cuproptosis was assessed. For *p* values < 0.001 and absolute correlation coefficients > 0.4, the corresponding lncRNAs were regarded as lncRNAs-related to cuproptosis. Afterwards, the limma package was used for differential expression analyses, and univariate Cox regression analysis was performed for screening prognostically-valuable (*p* < 0.05) and differentially expressed cuproptosis-associated lncRNAs.

### 4.3. Construction and Verification of a Prognostic Gene Signature

A least absolute shrinkage, and selection operator (LASSO)-Cox regression analysis was conducted to minimize the overfitting of the established model utilizing the R package, “glmnet” [49], and the covariances in 15 variables were eliminated. Afterwards, the multi-lncRNA prognostic profile associated with cuproptosis was established by multivariate Cox proportional hazards regression analysis [50]. The following equation was utilized for computing the risk scores: risk score = βlncRNA (1) × lncRNA (1) expression+ βlncRNA (2) × lncRNA (2) expression+ … + βlncRNA (n) × lncRNA (n), where β represents the coefficient of each lncRNA associated with cuproptosis acquired from Cox analysis. TCGA-PAAD was randomly classified into two groups at a ratio of 1:1, designated as test and training sets (Appendix A), and next, the patients were classified into high- and low-risk subgroups based on the median risk score. Scatter plots and risk curves were visualized using the R 4.2.0 software (https://www.r-project.org/) to reveal survival status related to the risk scores of PAAD patients. The difference in total survival between the two groups was compared using the Kaplan–Meier log-rank test. The R packages, “survival” [51], “survminer” [52], “survival ROC” [53], and “time ROC” [54], were used to plot receiver operating characteristic curves (ROC) corresponding to 1-, 2-, and 3-year survival based on clinicopathological features (containing age, stage, sex, and grade) and risk scores for predicting prognoses. The “Rtsne” [55] and “ggplot 2” [56] packages in R were employed for the principal component analysis (PCA), predicated on the characteristic expression of lncRNAs associated with cuproptosis, to distinguish between high- and low-risk PAAD patients. Wilcoxon test was utilized for the comparison of differences in survival between PAAD patients in high- and low-risk groups with different clinicopathological characteristics. The R package, “maftools” [57], was used for calculating and visualizing the mutation data in PAAD samples, including missense, frameshift indels, nonsense, and frameshift indels. Tumor mutation burden (TMB) was calculated using tumor-specific mutated genes.

### 4.4. Stemness Index Data Analysis

DNA dryness score (DNAss) based on the DNA methylation modes and RNA dryness score (RNAs) according to the mRNA expression. From the UCSC Xena (http://xena.ucsc.edu/, accessed on 31 March 2022) database, the stemness index data were downloaded, and the outcomes were visualized using the “limma” [58] and “corrplot” [59] packages in R.

### 4.5. Tumor Microenvironment and Clinical Treatment Response Analysis Using the Prognostic Risk Signature

For TCGA-PAAD, a single-sample gene set enrichment analysis (ssGSEA) was performed for quantifying the relative level of infiltration of 28 immune cell types in the TME, and the Wilcoxon rank-sum test was employed to analyze the differences in the abundances of immune cells between the two risk groups. Moreover, the correlation between immune cells and risk features based on a variety of currently accepted approaches, including QUANTISEQ, TIMER, XCELL, EPIC, MCPOUNTER, CIBERSORT, and CIBERSORT-ABS, were assessed. Spearman’s correlation analysis was employed to evaluate the immune infiltrates and the risk score correlations. Eventually, we evaluated whether there were differences in immune checkpoint gene expression between the two groups using “ggpubr” [60]. The “pRRophetic” [61] package in R was employed for drug IC_50_ prediction based on an online tool for cancer susceptibility genomics (GDSC).

### 4.6. Gene Set Enrichment Analysis (GSEA) and Functional Enrichment

Using GSEA (utilizing GSEA 4.1.0) [62], the major enriched cell signaling pathways between the two groups were identified based on the developed cuproptosis-related lncRNA signature. Statistical significance was defined based on FDR < 0.25 and *p* < 0.05. Furthermore, Kyoto Encyclopedia of Genes and Genomes (KEGG) enrichment analysis was conducted to obtain significant signaling pathways involved in the process.

### 4.7. Validation of Bioinformatics Results by qRT-PCR

From the Second Affiliated Hospital of Nanchang University, 32 pairs of PC tissues and corresponding neighboring pancreatic tissues were acquired between 2019 and 2021. The research protocol with human participants was reviewed and authorized by the Ethics Committee of the Cancer Hospital, Chinese Academy of Medical Sciences and the Second Affiliated Hospital of Nanchang University. From the tissues, the total RNA was extracted using the TRIzol reagent (Invitrogen, Waltham, MA, USA) following the manufacturer’s protocol. The cDNA was synthesized by reverse transcription using PrimeScript RT MasterMix (Takara Biomedical Technology (Beijing) Co., Ltd., Beijing, China). Relative RNA levels were determined using the SYBR Green PCR Mastermix (Takara Biomedical Technology (Beijing) Co., Ltd.). Appendix A shows the primer sequences used for qRT-PCR in this study. The target lncRNA expression was standardized against that of GAPDH.

### 4.8. Subcutaneous Xenograft Model

A subcutaneous xenograft mouse model was used to evaluate the tumor-forming ability of PANC-1 cells with stably knocked-down LINC00853. Male BALB/c nude mice that were 4 weeks old were purchased from Hunan STA Laboratory Animal Co., Ltd., Changsha, China. For in vivo signal detection, the mice were anesthetized with isofluorane and imaged using a Lumina Series III IVIS (In Vivo Imaging System) instrument (PerkinElmer, MA, USA). IVIS Lumina demonstrated GFP expression in the BALB/c nude mice. All animal experiments were approved by the Laboratory Animal Science Center of Wuhan University and Chinese Academy of Medical Sciences/Peking Union Medical College.

### 4.9. Identification of Extracellular Acidification Rate (ECAR) and Oxygen Consumption Rate (OCR)

Cellular glycolysis and cellular mitochondrial respiration levels were assessed using the XF Cell Mito stress test kit and Glycolysis Stress Test Kit (Seahorse Bioscience), respectively, following the manufacturer’s instructions on the Extracellular Flux Analyzer XF96 (Seahorse Bioscience, Billerica, MA, USA). Cells were seeded in an XF cell culture plate to reach 90% confluence, which were seeded at 12,000 cells/well. Directly before the assays, these cells were changed from a culture medium to an assay medium and incubated for 1 h at 37 °C. After baseline measurements, various chemicals prepared in the assay medium were sequentially injected into each well and subjected to measurement of ECAR or OCR respectively.

### 4.10. Western Blotting

Western blotting was used to determine the expression of PFKFB3 in si-LINC00853 BxPC-3 cells and si-LINC00853 PANC-1 cells. We used primary antibodies against PFKFB3 (1:1000; ab181861, Abcam, MA, USA) and GADPH (1:5000; Cat., No. 10494-1-AP; Proteintech, Wuhan, China). The membranes were washed three times with tris-buffered saline containing 0.1% Tween-20 at 4 °C and incubated with an HRP-labelled goat anti-rabbit IgG secondary antibody (1:5000; Cat. No. ab6728; Abcam). The protein bands were visualised and analysed using an ECL system and the Image J software (https://imagej.net/), respectively.

### 4.11. Statistical Analysis

Data processing, statistical analyses, and plotting of graphs were conducted using the GraphPad Prism 8 software (GraphPad Software, San Diego, CA, USA) and R 4.0.3. The χ^2^ test and Wilcoxon rank-sum test were utilized for the comparison of categorical and continuous variables between the two groups, respectively. Differences in progression-free survival (PFS) and overall survival (OS) between the two groups were identified by the log-rank test following the Kaplan–Meier approach. Statistical significance was regarded at *p* < 0.05.

## Figures and Tables

**Figure 1 ijms-24-07923-f001:**
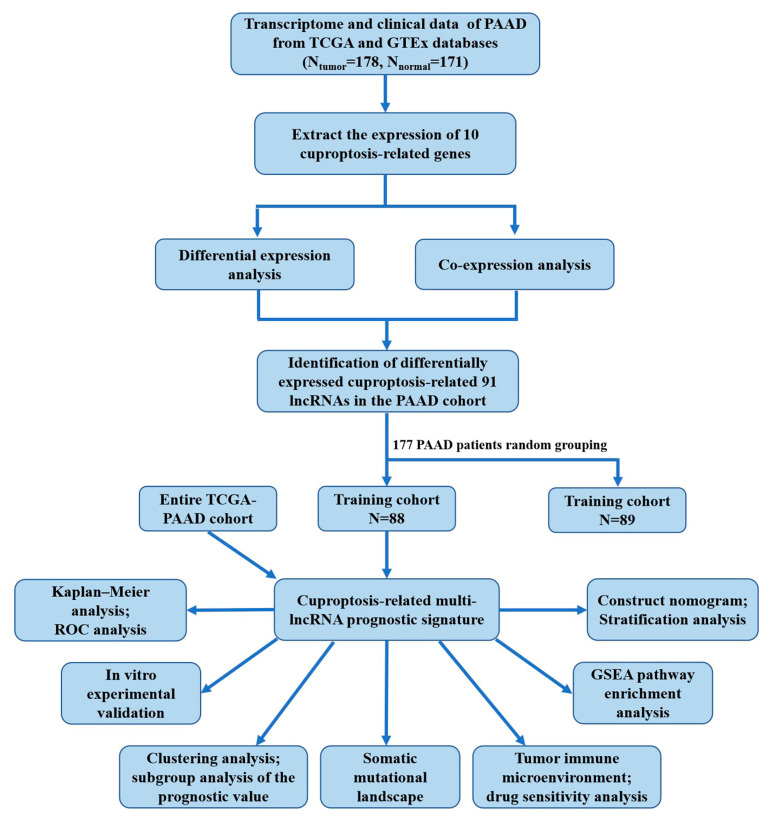
Flowchart of the study design.

**Figure 2 ijms-24-07923-f002:**
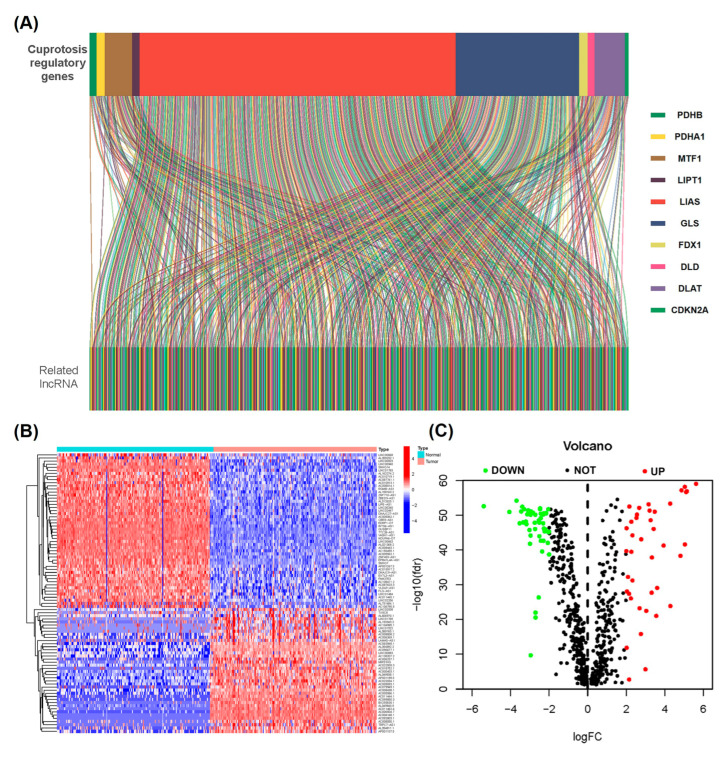
Selecting cuproptosis-associated lncRNAs in patients with PC. (**A**) The Sankey diagram for the network of lncRNAs and cuproptosis-associated genes. (**B**) The heatmap volcano plot together with (**C**) volcano plot of the heatmap of the 91 differentially expressed lncRNAs related to cuproptosis.

**Figure 3 ijms-24-07923-f003:**
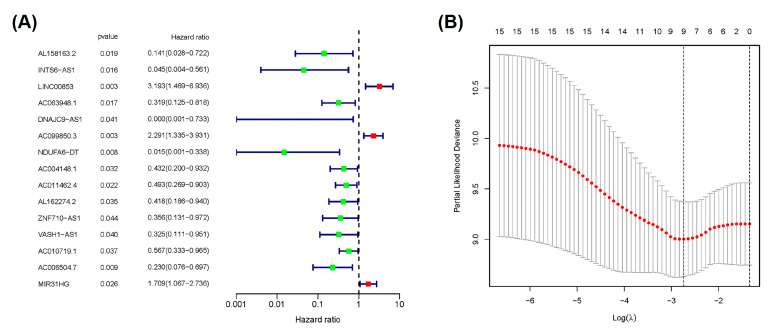
Establishment of cuproptosis-associated lncRNAs signature. (**A**) The forest plot of 15 cuproptosis-associated lncRNAs related to the overall survival of patients with PAAD following univariate Cox regression analysis. (**B**) Distribution of partial likelihood deviations by LASSO regression.

**Figure 4 ijms-24-07923-f004:**
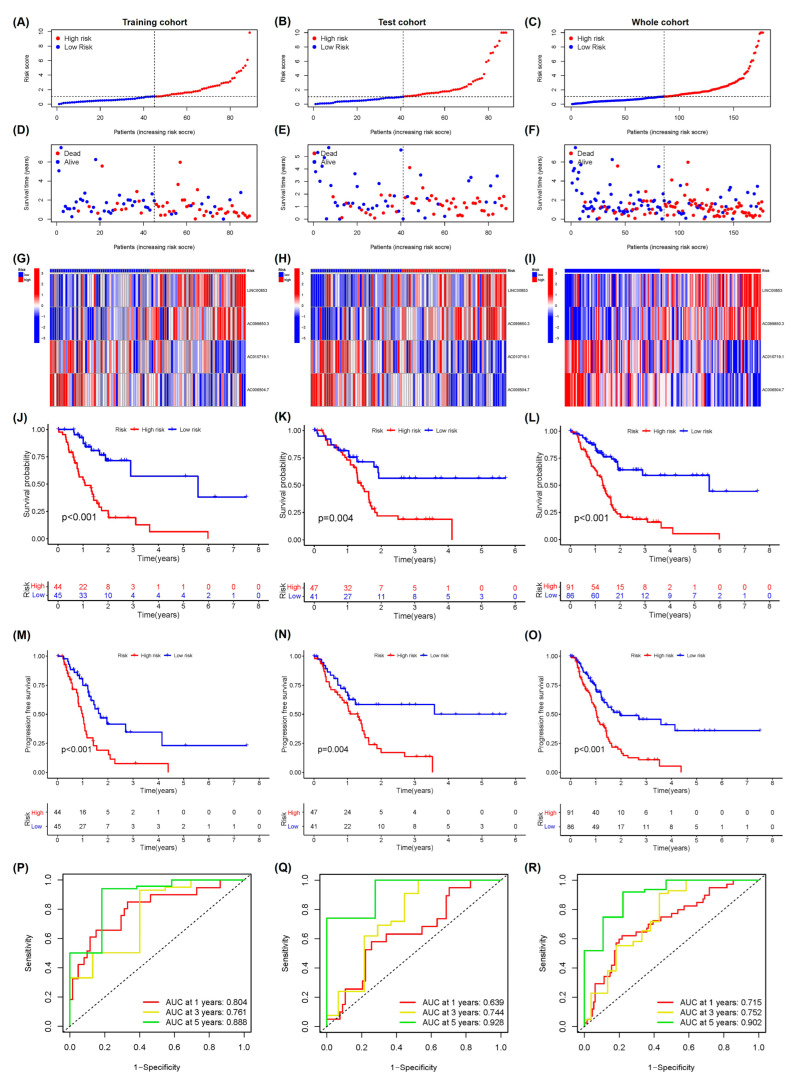
Assessment and verification of cuproptosis-associated lncRNA signature for its prognosis value in PAAD patients in test, training, and whole cohorts. The risk score distribution in the training (**A**), test (**B**), and whole (**C**) cohorts. The distributions of risk score, overall survival, and overall survival status in the training (**D**), test (**E**), and whole (**F**) cohorts. The heatmap revealed the expression of four cuproptosis-associated lncRNAs in the training (**G**), test (**H**), and whole (**I**) cohorts. The Kaplan–Meier survival curve of PFS and OS in the high- and low-risk groups in the training (**J,M**), test (**K**,**N**), and whole (**L**,**O**) cohorts. The analyses of ROC in the training (**P**), test (**Q**), and whole (**R**) cohorts.

**Figure 5 ijms-24-07923-f005:**
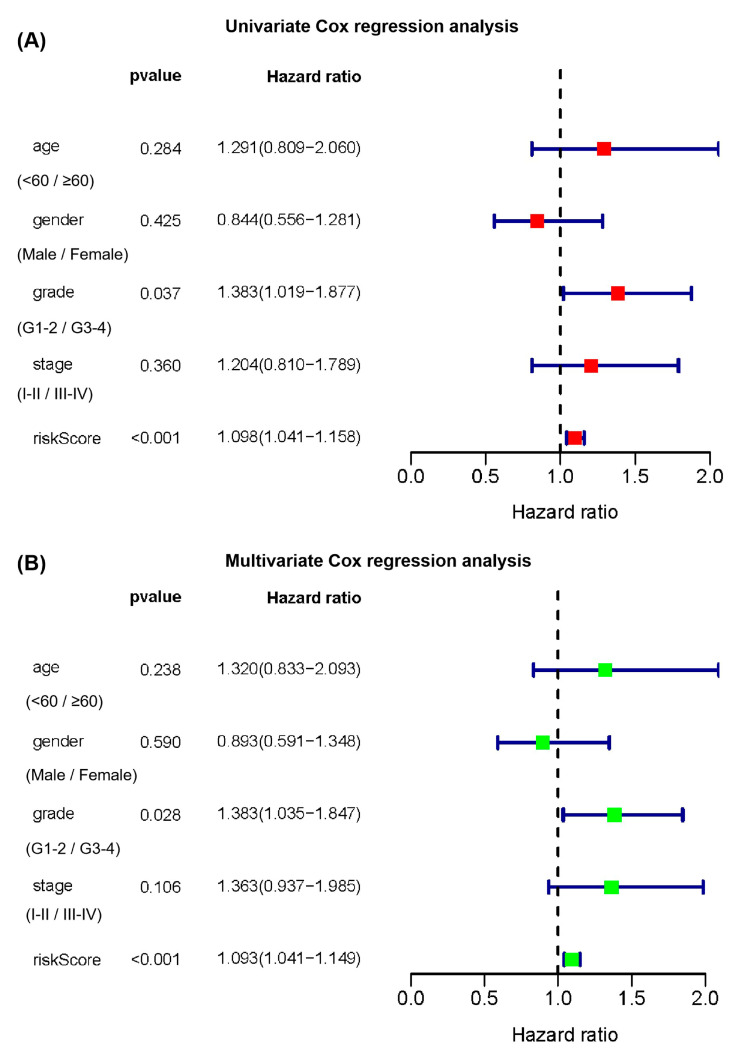
Independent analysis of the cuproptosis-related multi-lncRNA signature. (**A**) Univariate Cox and (**B**) multivariate Cox regression analyses for the signature and clinical features.

**Figure 6 ijms-24-07923-f006:**
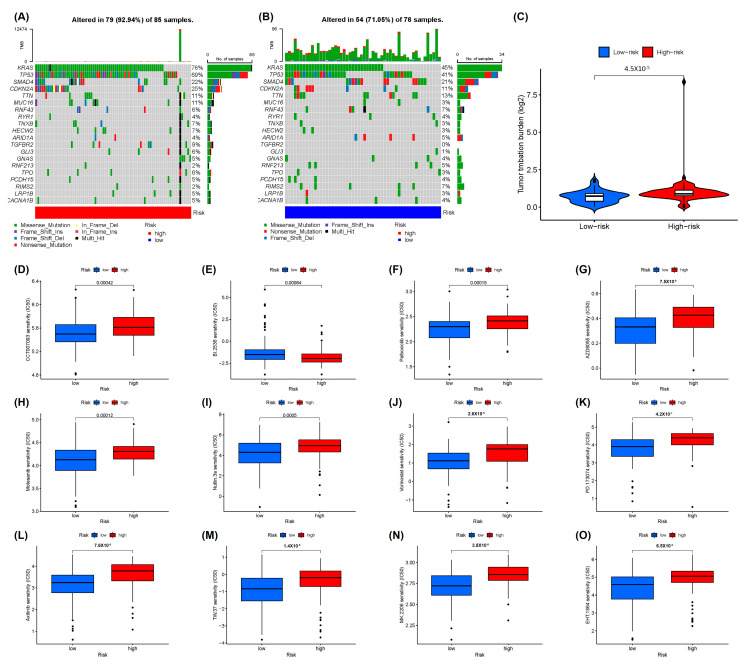
Drug sensitivity analysis and somatic mutational landscape according to the cuproptosis-associated lncRNA signature. (**A**,**B**) Waterfall plots of the top 20 mutated genes in the high- and low-risk groups in the whole cohort. (**C**) The difference of tumor mutational burden between the high- and low-risk groups in entire cohort. Drug sensitivity analysis for (**D**) WIP1 inhibitor (CCT007093), (**E**) PLK inhibitor (BI. 2536), (**F**) palbociclib, (**G**) mTOR inhibitor (AZD8055), (**H**) motesanib, (**I**) MDM2 inhibitor (Nutlin.3a), (**J**) HDAC inhibitor (Vorinostat), (**K**) FGF-VEGF receptor tyrosine kinase inhibitor (PD.173074), (**L**) axitinib, (**M**) Bcl2 inhibitor (TW.37), (**N**) AKT inhibitor (MK.2206), and (**O**) RAC family small GTPase inhibitor (EHT.1864).

**Figure 7 ijms-24-07923-f007:**
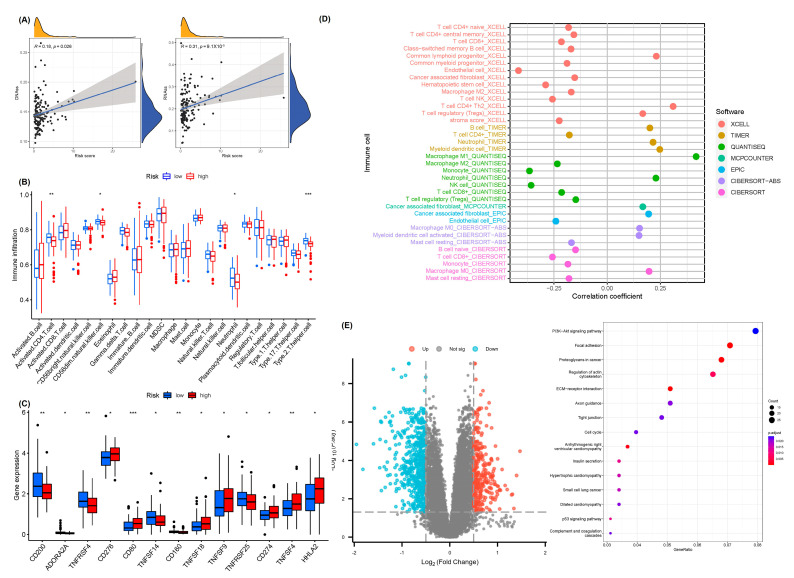
Correlation analysis between tumor immune microenvironment and cuproptosis-related lncRNA prognostic signature. (**A**) The correlation between RNAss, DNAss, and risk score. (**B**) Comparisons of abundances of immune cells in the tumor microenvironment between both risk groups by ssGSEA. (**C**) The immune checkpoint expression between risk groups. * *p* < 0.05, ** *p* < 0.01, *** *p* < 0.001. (**D**) The immune cell bubble between risk groups. (**E**) Differentially expressed genes and gene set enrichment analysis for the enriched underlying biological pathways between risk groups.

**Figure 8 ijms-24-07923-f008:**
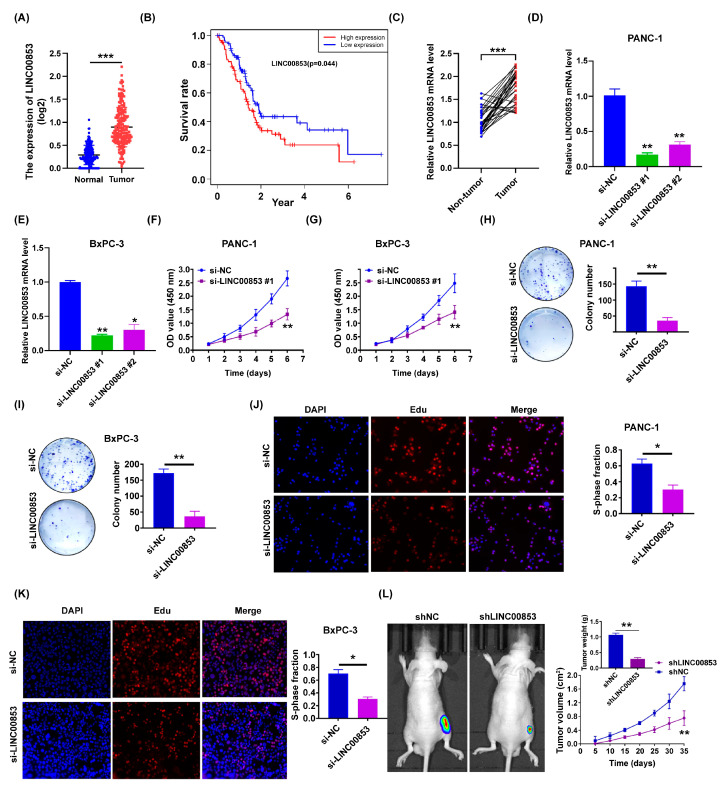
Overexpression of LINC00853 in PC and LINC00853 promotes pancreatic cancer cell growth in vivo and in vitro. (**A**) The expression profile of LINC00853 in TCGA-PAAD dataset. *** *p* < 0.001. (**B**) Kaplan–Meier curves for predicting the overall survival for both groups of patients with pancreatic cancer classified by high- and low LINC00853 expression in TCGA-PAAD dataset. (**C**) LINC00853 expression in the clinical pancreatic cancer tissues compared to the neighboring non-carcinoma normal tissues according to qRT-PCR analysis (*n* = 32). *** *p* < 0.001. (**D**,**E**) qRT-PCR was performed for determining the expression of LINC00853 in BxPC-3 and PANC-1 cells transfected with si-LINC00853. * *p* < 0.05, ** *p* < 0.01, *n* = 3. (**F**,**G**) CCK-8 analysis revealed pancreatic cancer cell proliferation following LINC00853 knockdown. ** *p* < 0.01, *n* = 3. (**H**,**I**) The quantification and representative images of colony formation assays for pancreatic cancer cells transfected with si-LINC00853. ** *p* < 0.01, *n* = 3. (**J**,**K**) The quantification and representative images of EdU assays for the pancreatic cancer cells transfected with si-LINC00853. * *p* < 0.05, *n* = 3. (**L**) shLINC00853/PANC-1 cells were injected subcutaneously into nude mice, and the tumor volumes were detected on the indicated dates; at the end of the experiment, tumors were dissected, weighed, and imaged. *n* = 6, ** *p*  <  0.01.

**Figure 9 ijms-24-07923-f009:**
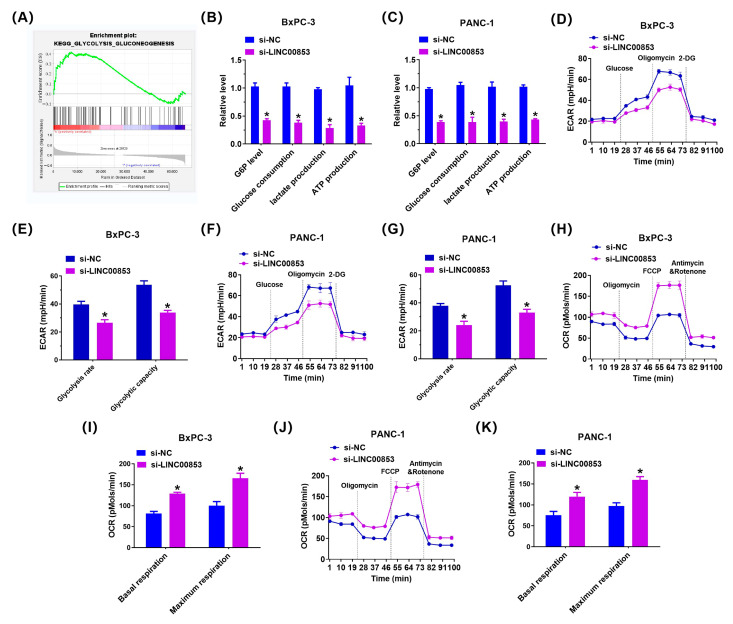
LINC00853 enhances aerobic glycolysis in pancreatic cancer cells. (**A**) GSEA comparing the gene sets of glycolysis targets in LINC00853 high PC patients. Data were obtained from TCGA database. (**B**,**C**) Cellular glucose consumption, G6P levels, ATP levels, and lactate generation in BxPC-3/si-LINC00853 and PANC-1/si-LINC00853. * *p* < 0.05, *n* = 3. (**D**–**G**) ECAR data showing the glycolytic rate and capacity in BxPC-3/si-LINC00853 and PANC-1/si-LINC00853. * *p* < 0.05, *n* = 3. (**H**–**K**) OCR results showing basal respiration and maximum respiration in BxPC-3/si-LINC00853 and PANC-1/si-LINC00853 cells. * *p* < 0.05, *n* = 3.

**Figure 10 ijms-24-07923-f010:**
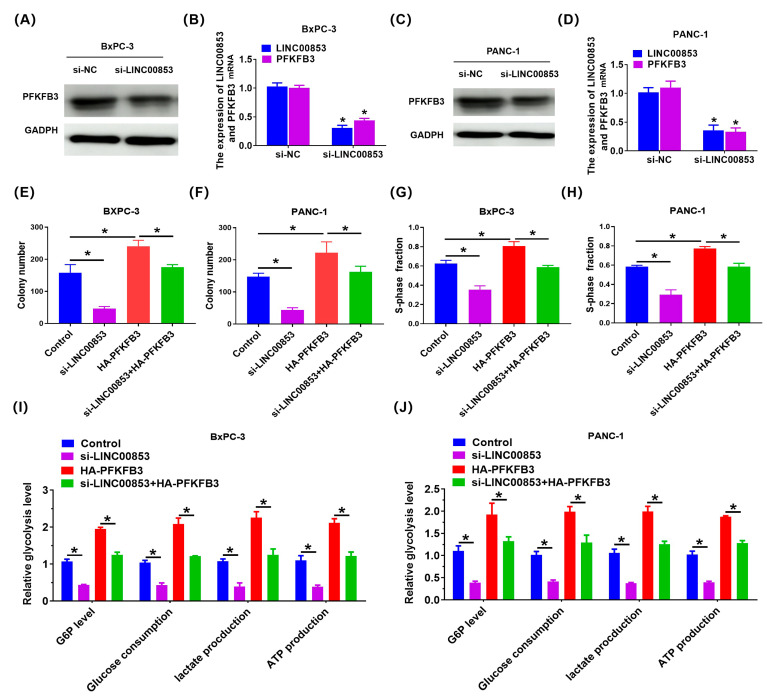
LINC00853 enhanced aerobic glycolysis and proliferation through PFKFB3 in PC cells. (**A**,**C**) Western blot was applied to measure PFKFB3 expression. (**B**,**D**) qRT-PCR was applied to measure LINC00853 and PFKFB3 expression. * *p* < 0.05, *n* = 3. (**E**–**H**) The proliferation ability was measured in the indicated groups. * *p* < 0.05. (**I**,**J**) Cellular glucose consumption, G6P levels, ATP levels, and lactate generation in the specific groups. * *p* < 0.05, *n* = 3.

**Table 1 ijms-24-07923-t001:** The factors in the risk signature.

lncRNA	Coefficient
LINC00853	1.33435227976852
AC099850.3	0.794073987427955
AC010719.1	−0.766590583753927
AC006504.7	−1.51899571329944

**Table 2 ijms-24-07923-t002:** Clinical information of pancreatic cancer patients in the training, validation, and entire cohort.

Variable	Group	Entire Cohort (*n* = 177)	Train Cohort (*n* = 88)	Test Cohort (*n* = 89)	*p*-Value
Age	≤65	91 (51.41%)	46 (52.27%)	45 (50.56%)	0.9384
	>65	86 (48.59%)	42 (47.73%)	44 (49.44%)	
Gender	Female	80 (45.2%)	41 (46.59%)	39 (43.82%)	0.8264
	Male	97 (54.8%)	47 (53.41%)	50 (56.18%)	
Grade	G1	28 (15.82%)	18 (20.45%)	10 (11.24%)	0.3202
	G2	93 (52.54%)	42 (47.73%)	51 (57.3%)	
	G3	51 (28.81%)	25 (28.41%)	26 (29.21%)	
	G4	3 (1.69%)	2 (2.27%)	1 (1.12%)	
	unknow	2 (1.13%)	1 (1.14%)	1 (1.12%)	
Stage	Stage I	20 (11.3%)	7 (7.95%)	13 (14.61%)	0.4895
	Stage II	145 (81.92%)	73 (82.95%)	72 (80.9%)	
	Stage III	2 (1.13%)	1 (1.14%)	1 (1.12%)	
	Stage IV	6 (3.39%)	4 (4.55%)	2 (2.25%)	
	unknow	4 (2.26%)	3 (3.41%)	1 (1.12%)	
T stage	T1	5 (2.82%)	2 (2.27%)	3 (3.37%)	0.3211
	T2	27 (15.25%)	9 (10.23%)	18 (20.22%)	
	T3	141 (79.66%)	74 (84.09%)	67 (75.28%)	
	T4	2 (1.13%)	1 (1.14%)	1 (1.12%)	
	unknow	2 (1.13%)	2 (2.27%)	0 (0%)	
M stage	M0	81 (45.76%)	40 (45.45%)	41 (46.07%)	0.6936
	M1	6 (3.39%)	4 (4.55%)	2 (2.25%)	
	unknow	90 (50.85%)	44 (50%)	46 (51.69%)	
N stage	N0	50 (28.25%)	20 (22.73%)	30 (33.71%)	0.172
	N1	121 (68.36%)	64 (72.73%)	57 (64.04%)	
	unknow	6 (3.39%)	4 (4.55%)	2 (2.25%)	

## Data Availability

The datasets ANALYZED for this study can be found at https://portal.gdc.cancer.gov/, accessed on 31 March 2022. The Cancer Genome Atlas (TCGA); https://dcc.icgc.org/releases/current/Projects, accessed on 31 March 2022. The International Cancer Genomics Consortium (ICGC, LIRI-JP; https://www.ncbi.nlm.nih.gov/geo/query/acc.cgi?acc=gse78220,GSE77820, accessed on 31 March 2022). The current study follows TCGA, ICGC, and GEO data access policies and publication guidelines.

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
