# Peer review of "Systemic Analyses of Cuproptosis-Related lncRNAs in Pancreatic Adenocarcinoma, with a Focus on the Molecular Mechanism of LINC00853"

_ijms, 2023, doi:10.3390/ijms24097923_

Round 1
Reviewer 1 Report
In the current study, the authors described the development of a scoring system using cuproptosis related lncRNAs to predict differences in PAAD prognoses and studied the molecular mechanism of one lncRNA, the LINC00853, in the regulation of cellular glycolysis and proliferation. I have few questions and suggestions, which I hope can help improve the quality of manuscript.
Overall, this study lack of novelty. The methods used in the current study was seen in several other studies (doi: 10.3389/fgene.2022.947551; doi: 10.3389/fgene.2022.938259; doi: 10.1038/s41598-022-25998-2) with very high similarity.
For the current study and the several other studies mentioned above, I doubt the reliability about the statement of “cuproptosis -associated lncRNAs”. The method described in the manuscript to identify lncRNAs is based on the “co-expression” of lncRNAs with “10 cuproptosis-realted genes” and “differential expression” of lncRNAs between normal and PD patients. There is no direct evidence to show that these lncRNAs are involved in cuproptosis. The validation between the identified lncRNAs and the “cuproptosis related genes” is needed to support that these lncRNAs are “cuproptosis -associated lncRNAs”.
Also, I would suggest the authors add more information regarding how each lncRNA related to the corresponding cuproptosis related genes, and whether PAAD affect the cuproptosis genes (differential of the cuproptosis genes between normal and PAAD patients?). For example, the target lncRNA LINC00853 was reported before as a potential HCC biomarker (DOI: 10.1002/1878-0261.12745). The authors need better evidence showing that these lncRNAs are really related to cuproptosis in PAAD.
The introduction about cuproptosis need to be revised. I suggest the authors add more information regarding the roles of cuproptosis in cancer progression, and molecular mechanisms of how cuproptosis get regulated. Also, in the lncRNA introduction part, add information about current knowledge in how lncRNAs may regulate cuproptosis.
In the discussion about molecular function of LINC00853, the author also did not test how this lncRNA may related to cuproptosis, which made it less convincing that this lncRNA is related to cuproptosis.
The authors identified the differentially expressed lncRNAs between normal and cancer patients, will it be more appropriate to use these lncRNAs for biomarkers of risk in developing caner, other than the biomarker of prognose of cancer?
The train cohort and test cohort look very similar, and I doubt whether it can really test the reliability of the proposed scoring system.
Author Response
Response to the comments of the reviewer
Reviewer # 1
We greatly appreciate the Reviewer’s excellent suggestions that are very helpful to improve the integrity and quality of our manuscript. As recommended, we have revised the manuscript accordingly. The point-by-point responses/revisions are as follows.
Comment 1: “In the current study, the authors described the development of a scoring system using cuproptosis related lncRNAs to predict differences in PAAD prognoses and studied the molecular mechanism of one lncRNA, the LINC00853, in the regulation of cellular glycolysis and proliferation. I have few questions and suggestions, which I hope can help improve the quality of manuscript. Overall, this study lack of novelty. The methods used in the current study was seen in several other studies (doi: 10.3389/fgene.2022.947551; doi: 10.3389/fgene.2022.938259; doi: 10.1038/s41598-022-25998-2) with very high similarity.”
Response:
Thanks for the time and suggestions of the reviewer. As the reviewer mentioned, the bioinformatics analysis method is partially similar to that of other previous tumor-related bioinformatic studies. However, the focus of this study is not to make innovations in analytical methods, but to explore the role of cuproptosis-related lncRNAs in pancreatic cancer. In fact, there has not been any study that has systematically analyzed cuproptosis-related lncRNAs in pancreatic cancer based on multiple datasets yet. More importantly, in this study, we explored the role of LINC00853 in the progression of pancreatic cancer through wet experiments, and these findings have not been reported before.
Comment 2: “For the current study and the several other studies mentioned above, I doubt the reliability about the statement of “cuproptosis -associated lncRNAs”. The method described in the manuscript to identify lncRNAs is based on the “co-expression” of lncRNAs with “10 cuproptosis-realted genes” and “differential expression” of lncRNAs between normal and PD patients. There is no direct evidence to show that these lncRNAs are involved in cuproptosis. The validation between the identified lncRNAs and the “cuproptosis related genes” is needed to support that these lncRNAs are “cuproptosis -associated lncRNAs.”
Response:
In this study, cuproptosis-related lncRNAs were obtained by correlation analysis between cuproptosis-related genes and differentially expressed lncRNAs in pancreatic cancer using Pearson's test. This method has been applied in a variety of published studies. For example, doi: 10.3389 / fgene. 2022.947551; Doi: 10.3389 / fgene. 2022.938259; doi: 10.1038/s41598-022-25998-2.
Comment 3: “Also, I would suggest the authors add more information regarding how each lncRNA related to the corresponding cuproptosis related genes, and whether PAAD affect the cuproptosis genes (differential of the cuproptosis genes between normal and PAAD patients?). For example, the target lncRNA LINC00853 was reported before as a potential HCC biomarker (DOI: 10.1002/1878-0261.12745). The authors need better evidence showing that these lncRNAs are really related to cuproptosis in PAAD.”
Response:
Thanks for the advice of the reviewers. Cuproptosis-related lncRNAs in this study were obtained using correlation analysis. The method has been applied in many published studies, and is applicable (doi: 10.3389 / fgene. 2022.947551; doi: 10.3389 / fgene. 2022.938259; doi:10.1038/s41598-022-25998-2). We showed the association between lncRNAs and these genes through the Sankey diagram. Also, we have uploaded the Supplementary Table 1 illustrating this. In addition, the differential expressions of the cuproptosis-related genes between normal and PAAD patients have been shown in Figure 2B-C.
Comment 4: “The introduction about cuproptosis need to be revised. I suggest the authors add more information regarding the roles of cuproptosis in cancer progression, and molecular mechanisms of how cuproptosis get regulated. Also, in the lncRNA introduction part, add information about current knowledge in how lncRNAs may regulate cuproptosis.”
Response:
Thank you for your suggestion. According to your suggestion, we have revised the introduction of cuproptosis and added the molecular mechanism of cuproptosis regulation. As for the current knowledge how lncRNAs regulate cuproptosis, the current mechanism is not clear, especially in PAAD, and this study is the first to explore it.
Comment 5: “In the discussion about molecular function of LINC00853, the author also did not test how this lncRNA may related to cuproptosis, which made it less convincing that this lncRNA is related to cuproptosis.”
Response: We thank the reviewer for raising this issue. According to the Reviewer’s suggestions. The association of this lncRNA and cuproptosis has been determined using Pearson's test and was shown in a Sankey diagram, as mentioned above.
Comment 6: “The authors identified the differentially expressed lncRNAs between normal and cancer patients, will it be more appropriate to use these lncRNAs for biomarkers of risk in developing caner, other than the biomarker of prognose of cancer?”
Response:
Thanks for your advice. While the lncRNAs are differentially expressed between normal and tumor tissues, they have also shown prognostic effects in pancreatic patients (Figure 3A). In this paper, our focus was on the latter and the former could be explored in future research. This has been acknowledged in our discussion.
Comment 7: “The train cohort and test cohort look very similar, and I doubt whether it can really test the reliability of the proposed scoring system.”
Response:
Thank you for the valuable question. When we establish the model, we divided the TCGA-PAAD cohort into a training cohort and test cohort at the ratio of 1:1, and performance of the scoring system was tested by KM survival analysis of the cohorts. The performance of the system is robust in the cohorts above. In addition, we uploaded the patients’ information of the training cohort and test cohort in the Supplemental Table 5.
Reviewer 2 Report
Chen et al., aims to identify cuproptosis-related lncRNAs involved in pathogenesis of pancreatic cancer (PC). To this end, the authors first establish a correlation between the expression patterns of cuproptosis-related mRNAs and lncRNAs. Of 672 such lncRNAs, four lncRNAs stand out that could divide patients into high- and low-risk groups. Interestingly, LINC00853 correlated well with the prognosis of the PC patients. In vitro studies in two differenct PC cell lines showed the oncogenic functions of this lncRNA. At the molecular level, LINC00853 expression correlated with glycolysis. Along this line, knockdown of LINC00853 reduced glycolysis and PFKFB3 protein expression. Chen et al also shows that the oncogenic function of LINC00853 could be partially rescued by PFKFB3 overexpression, suggesting the presence of a LINC00853-PFKFB3 axis that could regulate aerobic glycolysis and proliferation in PC cells.
I believe that Chen et al presents a novel axis, the LINC00853-PFKFB3 axis, in the pathogenesis of PC that could be of interest to the scientists working in this field. The manuscript is well-written and appears to be cohesive. I suggest the following points for consideration to improve the manuscript.
Major points:
1. The details of methods are missing for most, if not all, protocols. Additionally, no references are provided for any of the methods described. For example, the reference for 10 genes in line 483 is missing. None of the bioinformatic tools used in the manuscript has any references. No details are provided for ECAR and OCR experiments. It is impossible for one to replicate any of the experiments in this manuscript.
2. I did see any subsection of western blotting in Materials and Methods although the authors present WB results.
3. The authors present only one replicate of WB results. Please provide the original images of all replicates. If there is only one replicate, please explain how the SD was obtained Fig 10B-D.
4. Please switch either the figures or the captions of panels B and C in Figure 2. They appear to be mislabeled.
5. Please avoid descriptive usage while refering to the data; rather, be quantitative (e.g., instead of “markedly enhanced” in line 274, please state the quantitive numbers).
6. Please state either in the Methods or in the figure caption, how many times each experiment was repeated.
Minor points:
1. Please avoid using the term “long-chain non-coding RNAs”. Instead, simply use “long non-coding RNAs”. There are sevearal such situations throughout the manuscript.
2. Line 30, “si-RNA” should be “siRNA”
Author Response
Response to the comments of the reviewers
Reviewer # 2
We greatly appreciate the reviewer’s time spent in critiquing our manuscript and the constructive suggestions that were offered. We have performed the recommended experiments and have revised the manuscript accordingly. The point-by-point responses/revisions are as follows:
(Reviewer Comments to the Author): “Chen et al., aims to identify cuproptosis-related lncRNAs involved in pathogenesis of pancreatic cancer (PC). To this end, the authors first establish a correlation between the expression patterns of cuproptosis-related mRNAs and lncRNAs. Of 672 such lncRNAs, four lncRNAs stand out that could divide patients into high- and low-risk groups. Interestingly, LINC00853 correlated well with the prognosis of the PC patients. In vitro studies in two differenct PC cell lines showed the oncogenic functions of this lncRNA. At the molecular level, LINC00853 expression correlated with glycolysis. Along this line, knockdown of LINC00853 reduced glycolysis and PFKFB3 protein expression. Chen et al also shows that the oncogenic function of LINC00853 could be partially rescued by PFKFB3 overexpression, suggesting the presence of a LINC00853-PFKFB3 axis that could regulate aerobic glycolysis and proliferation in PC cells.
I believe that Chen et al presents a novel axis, the LINC00853-PFKFB3 axis, in the pathogenesis of PC that could be of interest to the scientists working in this field. The manuscript is well-written and appears to be cohesive. I suggest the following points for consideration to improve the manuscript.”
Major points:
Comment 1: “The details of methods are missing for most, if not all, protocols. Additionally, no references are provided for any of the methods described. For example, the reference for 10 genes in line 483 is missing. None of the bioinformatic tools used in the manuscript has any references. No details are provided for ECAR and OCR experiments. It is impossible for one to replicate any of the experiments in this manuscript.”
Response: We thank the reviewer for pointing out this important issue. According to the reviewer’s suggestion, we have added the relevant references and methods of the experiments in this manuscript.
Comment 2: “I did see any subsection of western blotting in Materials and Methods although the authors present WB results.”
Response: We thank the reviewer for the suggestion, which will be very helpful to our article. As the reviewer suggested, we have added the description of western blotting process in this manuscript.
Comment 3: “The authors present only one replicate of WB results. Please provide the original images of all replicates. If there is only one replicate, please explain how the SD was obtained Fig 10B-D.”
Response: We apologize for causing this misunderstanding. In Figure 10B and 10D, we detected the levels of LINC00853 and PFKFB3 mRNA by qRT-PCR, rather than the gray value of the blots. We have made modifications in the new Figure 10B-D accordingly.
Comment 4: “Please switch either the figures or the captions of panels B and C in Figure 2. They appear to be mislabeled.”
Response: We apologize for this mislabeling. According to reviewer suggestions, we have made corresponding corrections.
Comment 5: “Please avoid descriptive usage while refering to the data; rather, be quantitative (e.g., instead of “markedly enhanced” in line 274, please state the quantitive numbers). ”
Response: We thank the reviewer for raising this issue. We have deleted this description to avoid vagueness.
Comment 6: “Please state either in the Methods or in the figure caption, how many times each experiment was repeated.”
Response: We thank the reviewer to raise this important issue. According to the Reviewer’s suggestions, we have made corrections accordingly in Methods and in figure legends.
Minor points:
Comment 1: “Please avoid using the term “long-chain non-coding RNAs”. Instead, simply use “long non-coding RNAs”. There are sevearal such situations throughout the manuscript.”
Response:We really appreciate the reviewer’s suggestion that will be helpful to improve the quality of our articles. According to reviewer suggestion, we have replaced all “Long-chain non-coding RNAs” with “Long non-coding RNAs”.
Comment 2:“Line 30, “si-RNA” should be “siRNA””
Response: According to reviewer suggestion, we have made corresponding corrections.
Reviewer 3 Report
Chen et al. elucidated the role of LINC00853 in pancreatic cancer from the perspective of Cuproptosis-related lnc RNA. They demonstrated the physiological role of LINC00853 in pancreatic cancer using existing large-scale database analyses and in vivo/in vitro experiments. While there may be some minor issues to be revised, overall, I believe it could be accepted after minor revisions.
2. Results
2.5 Analysis of tumor immune microenvironment and pathways related to prognostic features
Page 9 Line 222-227; This sentence describes the methods of analysis of DNA dryness score and RNA dryness score. I would recommend that this sentence should be included in the method section.
2.6. Independent analysis of the cuproptosis-related multi-lncRNA signature
Page 11 Line 257-267; This section is showing the result of multivariate analysis whether cuproptosis-related multiple lncRNA were independent of the existing clinicopathological indicators. It seems that this section is related to section 2.3 Subgroup analysis of the prognostic value of the cuproptosis-related multi-lncRNA signature. I would recommend that this section should be moved after the section 2.3.
Author Response
Response to the comments of the reviewers and revisions
Reviewer # 3
We greatly appreciate the reviewer’s time in reviewing our manuscript and the constructive suggestions offered here. We have performed the recommended experiments and have revised the manuscript accordingly. The point-by-point responses/revisions are as follows:
(Reviewer Comments to the Author): “Chen et al. elucidated the role of LINC00853 in pancreatic cancer from the perspective of Cuproptosis-related lnc RNA. They demonstrated the physiological role of LINC00853 in pancreatic cancer using existing large-scale database analyses and in vivo/in vitro experiments. While there may be some minor issues to be revised, overall, I believe it could be accepted after minor revisions.”
Comment 1: “ Analysis of tumor immune microenvironment and pathways related to prognostic features
Page 9 Line 222-227; This sentence describes the methods of analysis of DNA dryness score and RNA dryness score. I would recommend that this sentence should be included in the method section.”
Response: We really appreciate the reviewer’s suggestion, that will be helpful to improve the quality of our articles. This sentence have been included in the method section and the related methodological steps have been described.
Comment 2: “2. Independent analysis of the cuproptosis-related multi-lncRNA signature
Page 11 Line 257-267; This section is showing the result of multivariate analysis whether cuproptosis-related multiple lncRNA were independent of the existing clinicopathological indicators. It seems that this section is related to section 2.3 Subgroup analysis of the prognostic value of the cuproptosis-related multi-lncRNA signature. I would recommend that this section should be moved after the section 2.3.”
Response: We thank the reviewer for the suggestion, which will be very helpful to our article. The message conveyed in the two parts are similar and the two parts have been merged.
Round 2
Reviewer 1 Report
I appreciate the authors to address my questions.
In additon I have few minor following suggestions:
The use of abbreviations need to be revised (for example: missing full name of CRGs).
"P" should be italic in the added content.
Change "RT-qPCR" to "qRT-PCR".
Author Response
Response to the comments of the reviewer
Reviewer # 1
We greatly appreciate the reviewer’s time spent critiquing our manuscript and the constructive suggestions that were offered. We have performed the recommended experiments and have revised the manuscript accordingly. The point-by-point responses/revisions are as follows:
Comment 1: “The use of abbreviations need to be revised (for example: missing full name of CRGs).”
Response:We really appreciate the reviewer’s suggestion, that will be helpful to improve the quality of our articles. According to reviewer suggestion, we had made corresponding corrections.
Comment 2:“"P" should be italic in the added content.”
Response:According to reviewer suggestion, we had made corresponding corrections.
Comment 3:“"RT-qPCR" to "qRT-PCR".”
Response: We thank the reviewer to raise this important issue. According to the Reviewer’s suggestions, , we had made corresponding corrections.
Reviewer 2 Report
The authors have nicely addressed all the points raised. The manuscript can be accepted in the present form.
Author Response
Response to the comments of the reviewer
Reviewer # 2
(Reviewer Comments to the Author):“The authors have nicely addressed all the points raised. The manuscript can be accepted in the present form.”
Response: We greatly appreciate the reviewer’s time spent critiquing our manuscript and the constructive suggestions that were offered.